# Functions and the Emerging Role of the Foetal Liver into Regenerative Medicine

**DOI:** 10.3390/cells8080914

**Published:** 2019-08-16

**Authors:** Antonella Giancotti, Marco Monti, Lorenzo Nevi, Samira Safarikia, Valentina D’Ambrosio, Roberto Brunelli, Cristina Pajno, Sara Corno, Violante Di Donato, Angela Musella, Michele Francesco Chiappetta, Daniela Bosco, Pierluigi Benedetti Panici, Domenico Alvaro, Vincenzo Cardinale

**Affiliations:** 1Department of Maternal and Child Health and Urologic Sciences, Umberto I Hospital, Sapienza University of Rome, 00161 Rome, Italy; 2Department of Translation and Precision Medicine, Sapienza University of Rome, 00185 Rome, Italy; 3Department of Pathological Anatomy and Cytodiagnostic, Sapienza University of Rome, 00185 Rome, Italy; 4Department of Medico-Surgical Sciences and Biotechnologies, Sapienza University of Rome-Polo Pontino, 00185 Rome, Italy

**Keywords:** foetal liver, foetal organoids, regenerative medicine, foetal liver embryogenesis, foetal stem cells

## Abstract

During foetal life, the liver plays the important roles of connection and transient hematopoietic function. Foetal liver cells develop in an environment called a hematopoietic stem cell niche composed of several cell types, where stem cells can proliferate and give rise to mature blood cells. Embryologically, at about the third week of gestation, the liver appears, and it grows rapidly from the fifth to 10th week under WNT/β-Catenin signaling pathway stimulation, which induces hepatic progenitor cells proliferation and differentiation into hepatocytes. Development of new strategies and identification of new cell sources should represent the main aim in liver regenerative medicine and cell therapy. Cells isolated from organs with endodermal origin, like the liver, bile ducts, and pancreas, could be preferable cell sources. Furthermore, stem cells isolated from these organs could be more susceptible to differentiate into mature liver cells after transplantation with respect to stem cells isolated from organs or tissues with a different embryological origin. The foetal liver possesses unique features given the co-existence of cells having endodermal and mesenchymal origin, and it could be highly available source candidate for regenerative medicine in both the liver and pancreas. Taking into account these advantages, the foetal liver can be the highest potential and available cell source for cell therapy regarding liver diseases and diabetes.

## 1. Foetal Liver Embryogenesis and Foetal Functions

### 1.1. Foetal Liver Functions

In adult life, the liver is the biggest gland in the body, with both endocrine and exocrine functions. The major exocrine secretion consists of the production of bile, while endocrine functions include the secretion of several hormones such as insulin-like growth factors, angiotensinogen, and thrombopoietin. The liver is also indispensable for glycogen storage, drug detoxification, control of metabolism, regulation of cholesterol synthesis and transport, urea metabolism, and secretion of a wide number of plasma proteins including albumin and apolipoproteins [1]. In the foetus, the liver has two important properties: cardiovascular, being a vascular connection between the developing placental vessels to the heart, and haemopoietic, as a special tissue where blood stem cells reside before bone marrow development.

### 1.2. Cardiovascular Function

In foetal life, the liver takes up a very large portion of the abdominal cavity. In this stage, it does not perform the traditional digestive and filtration functions because nutrients are normally carried to the foetus from the mother via the placenta. Blood flow, rich in nutrients, arrives from the placenta to the foetus by the umbilical vein. It enters the abdomen at the umbilicus, passing upward along the free margin of the falciform ligament of the liver to its inferior surface. Blood flow in the umbilical vein has an oxygen saturation of approximately 80% compared with the 98% of the arterial adult circulation [2]. The ductus venosus, a little vein situated in the liver, carries some of this blood directly to the inferior vena cava, bypassing the liver circulation, where it joins blood from the lower trunk and extremities and from the liver. This flow arrives in the right atrium but is mostly diverted directly to the left atrium via a patent foramen oval in the atrial septum. From here, blood arrives to the left ventricle and ascending aorta to supply the coronary and cranial vascular beds [3,4]. Unlike in postnatal life, where the lungs receive the entire cardiac output, in foetal life, the collapsed lungs provide only 10% of the cardiac output [2]. In the presence of diminished nutrition or oxygenation, a greater volume of flow from the umbilical vein bypasses the liver and perfuses the head and neck of the foetus, privileging oxygen and nutrient delivery to the brain as part of the stated ‘brain-sparing’ effect [5,6]. In growth-restricted human foetuses, such brain-sparing responses are associated with cerebral vasodilation (measured using Doppler ultrasound as a low pulsatility index in the middle cerebral artery [3]) and with greater ductus venosus shunting [5,6]. Some of these adaptive changes seem to also occur in normally growing fetuses in late gestation; a large variation in the proportion of placental blood perfusing or bypassing the liver have been described in in such foetuses [7].

### 1.3. Haematopoiesis

The location of hematopoiesis during gestation varies from the first weeks until birth [8]. The first site involved is the yolk sac, where primitive hematopoiesis starts around seven weeks of gestational age [9], then is temporarily shared between the liver, spleen, and thymus, before finally being establish definitively in the bone marrow (Figure 1).

At 10 weeks gestational age, hematopoietic progenitor cells gradually migrate from the aorta-mesonephros-gonad region to colonize the liver, which becomes the major hematopoietic organ [1]. Significantly higher numbers of total erythroid precursors and different erythroid components (pronormoblasts, basophilic, and polychromatic normoblasts) are seen in the liver compared with the spleen and thymus. By fully supporting hematopoiesis, cells in the foetal liver constitute a hematopoietic stem cell (HSC) niche, an environment where stem cells can proliferate and give origin to mature blood cells, as happens during adult life in the bone marrow [10]. At 13 weeks and five days, the liver is colonized by 1000 Pluripotent Hematopoietic stem cells (pHSCs), 10,000 multipotent myeloid/lymphoid progenitors (MPPs), and their progeny, all expressing different markers such as c-kit, Sca-1, CD150, and CD. A complex of cell types assembles the hematopoietic microenvironment of the foetal liver. They include epitheliocytes, resident macrophages, and several stromal cell populations of mesenchymal origin such as hepatic stellate cells, fibroblasts, myofibroblasts, vascular smooth muscle and endothelial cells, and mesenchymal stromal cells (MSCs) [11].

Liver hematopoiesis in the prenatal period can be subdivided in four stages. It gradually rises in Stages I and II, reaches a peak in Stage III, and slowly regresses through Stage IV. This is consistent with the fact that primitive hematopoiesis is localized in the yolk sac during the first weeks of development and later on appears in the liver. The first signs of hematopoiesis in the human foetal liver are reported as soon as 5–8 weeks [12]. During Stage I, hematopoietic cells in the liver can mostly be pluripotent stem cells [13], which we observed as CD34þ cells mainly located in the sinusoids. These cells then migrate toward the hepatoblasts cords, where they create hematopoietic foci. During migration, these cells likely make contact with sinusoidal endothelial cells, which have shown an ability to support hematopoiesis [14,15]. During Stage II, at the interface between hepatoblasts cords and the sinusoidal lumen, in close contact with endothelial cells, hematopoietic foci have been observed. This agrees with the theory that sinusoidal endothelial cells could support hematopoiesis.

During Stage III, in the foetal liver, an increase in hematopoiesis activity in notiedas hematopoietic cells occupy at least 70% of the liver parenchyma. The greatest part of the cells comprising the hematopoietic foci are cells from the erythrocytic lineage, but a significant number of mature granulocytes is present, suggesting the presence of cells of myeloid lineage still in an undifferentiated state.

Stage IV is characterized by a reduction in hematopoietic activity. In this phase, hematopoietic cells occupy less than 30% of liver parenchyma.

In general, the erythropoietic lineage was the most representative lineage in terms of quantity in every stage, and its quantity was at least three times higher than myelomonocytic, suggesting how the liver could be mainly responsible for erythropoiesis [16].

Megakaryopoietic activity is described in the foetal liver but in scarcely represented. These cells are always solitary and are mostly located between hepatoblasts [17,18].

Hematopoietic stem cells are pluripotent and can form any hematopoietic cell. Their first step in intrahepatic maturation is to commit to a more limited range of lineage options, typically an erythromyeloid precursor or a common myelolymphoid progenitor [19,20,21].

Proerythroblast is the earliest morphologically identifiable erythroid precursor, which differentiates sequentially into an erythroblast, which can then enucleate to form a reticulocyte. The precursors show a gradual reduction in cell and nuclear size. On the other hand, an increase in hemoglobin concentrations is registered [22]. Liver-derived myelolymphoid progenitors subsequently develop into bipotent cells (B lymphocyte and myeloid or T lymphocyte and myeloid) before committing to produce a single cell lineage [19]. Some T lymphocyte progenitors have a bipotent commitment to the NK cell lineage as well [23]. The T lymphocyte precursors destined for transfer to the thymus are produced even in athymic mice [19,24], indicating that the foetal liver may play a role in promoting early T lymphocyte differentiation prior to releasing any precursor cells to the thymus and bone marrow [25]. Beyond 11 weeks and five days, erythroids such as lymphoid cells become increasingly more differentiated. T lymphocytes are the primary lymphoid cell progenitor present at 12 weeks and five days and then decline in number as development proceeds. In fact, at 14 weeks and five days, B lymphocyte progenitors forms the primarily lymphoid population due to a switch in the foetal liver.

Also, a granulocyte population develops throughout gestation. Granulocyte progenitors can be found in low numbers scattered throughout the liver parenchyma between 12 weeks and five days and 15 weeks and five days, then they clustered in small foci throughout of the liver. These foci get larger as development proceeds.

With advancing gestational age, a change in hematopoiesis sites are encountered. In particular, from about 13 weeks, T lymphocytes shift from the liver to the thymus. At about 15 weeks, erythroid and myelolymphoid precursors move to spleen and the and finally to bone marrow (16 weeks) [9,26], which is the definitive site where hematopoietic stem cells and progenitors of the adult hematopoietic system reside [9,27,28].

### 1.4. Foetal Liver Embryology

Embryologically liver, gallbladder, and biliary ducts system appears at about the third week of gestation as a ventral outgrowth of the endodermal epithelium (epithelial lining of the alimentary canal) at the distal end of the foregut (Cephalic portion of the primary intestine) [29]. The embryonic draft of the liver is called hepatic diverticulum. The Wnt/b-catenin signaling pathway plays a key role in the process of cells proliferation and differentiation of the primary intestine. This process includes the proliferation and differentiation of the hepatic progenitor cells to form hepatocytes.

It has been hypothesized that the origin of both hepatic diverticulum and ventral bud of the pancreas is derived from two cell populations of the embryonic endodermis. Sufficient levels of fibroblast growth factors (FGFs) secreted by the developing foetal cardiac mesoderm interact with bipotential cells, inducing hepatic diverticulum formation [30].

A rapid hepatic cell proliferation causes the extension of hepatic diverticulum into the septum transversum, a mass of splanchnic mesoderm separating the pericardial and peritoneal cavities. The septum trasversum forms the ventral mesogastrium in this region. The hepatic diverticulum expands quickly and divides into right and left branches [31]. Each branch gives rise to clusters of liver cells forming two solid masses that later give rise to the right and left lobes of the liver.

Meanwhile the liver cells continue their penetration into the septum trasversum, the link between hepatic diverticulum and the anterior gut narrows, thus forming the biliary pathway. The hepatic diverticulum has a cranial part representing the primordium of the liver. The caudal part becomes the primordium of the gallbladder. The proliferation of endodermal cells causes the formation of hepatocyte cords. These give rise to the epithelial lining of the intrahepatic part of the bile apparatus. The hepatic cords anastomose at the endothelium-lined spaces, the primordia of the hepatic sinusoids. The role of vascular endothelial growth factor signaling seems to be important in the development of the hepatic sinusoids. The formations of the bile by hepatic cells starts during the 12th week above the small caudal part of the hepatic diverticulum gives rise to the gallbladder, and the stalk of the diverticulum forms the cystic ducts. Originally, epithelial cells occlude the extrahepatic biliary apparatus but later canalize by means of vacuolization mechanism resulting from degeneration of these cells. The part of the diverticulum connecting the hepatic and cystic ducts to the duodenum becomes the bile duct. At first, this duct is connected to the ventral part of the duodenal loop. However, as a result of duodenum growth and rotation, the entrance of the bile duct is carried to the dorsal part of the duodenum. The fibrous hematopoietic tissue and Kupffer cells of the liver are derived from mesenchyme in the septum transversum. The liver grows rapidly from the fifth to the 10th weeks and fills a large part of the upper abdominal cavity representing approximately 10% of the total body weight in the 10th week of development. Liver development and functional segmentation are determined by the amount of oxygenated blood flowing from the umbilical vein to the liver. Initially, the right and left lobes are approximately the same size, but the right lobe soon becomes larger. Thus, developmental change in the liver is possibly related to its function. Glycogen is not present in the liver of young foetuses, but it appears at about the 30th week and gradually increases in amount [32]. Hepatic hemopoiesis consist in genesis and development of various types of blood cells. It starts in the foetal liver during the sixth week of gestation, reaches its maximal activity toward the sixth to seventh month, and then regresses [33]. Histologically, the liver is complete by the eighth to ninth month of gestation [34]. The liver of the foetus may have an accelerated growth both in morphology and function after approximately 32 weeks of gestation.

The foetal umbilical portal venous system forms in the embryo between the fourth and 10th week and is derived from two distinct embryologic precursors—vitelline veins originating from the yolk sac and the umbilical veins originating from the placenta [35]. These two distinct extraembryonic venous systems drain into the sinus venosus of the foetal heart [8].

Vitelline veins are paired structures connected by three anastomotic channels interrupted by developing hepatic trabeculae. As the embryo is folded laterally, the vitelline veins assume the drainage of the gut and the developing hepatic primordium. These veins are largely incorporated into the developing liver as hepatic sinusoids; only some parts of caudal and cranial vitelline segments persist as separate structures. As the left vitelline vein regresses, the cranial portion of the right vitelline vein widens to form the right hepato-cardiac channel. This forms the intrahepatic portion of the inferior vena cava (IVC) and contributes to the formation of ductus venosus. The ductus venosus allows placental blood to pass from the single left umbilical vein through the portal and then into the IVC and the right atrium. The caudal portions of the vitelline vein merge to form the portal vein. The umbilical vein system is formed by the division of the pre-hepatic trunk into two. One branch goes directly inside the hepatic parenchyma, and the other runs along the liver to open into the sinus venosus of the foetal heart [36].

The left umbilical vein also contributes to a portion of the left portal vein. The hepatic artery is visible for the first time at the eighth gestational week as a brunch of celiac trunk. Its development is dependent on the advanced intrahepatic portal system. After birth and following separation from the placental flow, umbilical circulation ceases, and the two umbilical arteries close first, followed by the umbilical vein. Blood to the liver through the umbilical vein is interrupted, and the ductus venosus closes.

### 1.5. Molecular Regulation of Liver Induction

The foregut endoderm has the potential to express liver-specific genes and to differentiate into liver tissue. However, this expression is blocked by inhibitory factors produced by surrounding tissues, including ectoderm, noncardiac mesoderm, and particularly the notochord. The cardiac mesoderm and endothelial cells that form the blood vessels secrete fibroblast growth factors 2 (FGF2) that block the inhibitory substances described above. In this way, the intestinal endodermis is instructed to express liver-specific genes. The septum trasversum produces bone morphogenic proteins (BMPs), which participate in this induction mechanism, increasing the endodermal response to the FGF2. Once this “instruction” is received, cells in the liver field differentiate into both hepatocytes and biliary cell lineages, a process that is at least partially regulated by hepatocyte nuclear transcription factors (HNF3 and HNF4).

### 1.6. Common Embryogenesis of Liver and Pancreas and Remnant Stem Cells in Adult Life

The liver, bile duct system, and pancreas share a common origin from the definitive ventral endoderm forming the foregut [37,38]. In human development, following the typical pattern of development of the so-called branching tissues, migrating cords of stem cells and alpha-fetoprotein positive hepatoblasts bulge into the mesenchyme of the septum transversum forming the ductal plate [37,39,40]. These anatomic structures undergo to a remodeling process that finally leads to the formation of intrahepatic bile ducts (BDs), comprising the finer branches—the canals of Hering (CoH). As far as the rest of the biliary tree, it is notable that the larger intrahepatic and extrahepatic BDs share a distinct common origin pattern. Indeed, the liver hilum, the larger intrahepatic BDs [39] originated from the elongation and branching of the common hepatic duct. The caudal part of the hepatic diverticulum harbours biliopancreatic stem/progenitors co-expressing SOX17 and PDX1 at approximately 28–32 embryonic days in humans (10.5 in mice) and gives rise to the extrahepatic biliary tree and ventral pancreas. Successively, specific transcription factors and signaling pathways will drive further commitment of pancreatic and biliary precursors [41] (Figure 2). The origin as branching tissues of liver, biliary tree, and pancreas from the foregut at the level of the future duodenum may be seen as the substrate of a proximal-to-distal longitudinal axis. Within larger intrahepatic and extrahepatic BDs, the peribiliary glands (PBGs) progressively bud from the epithelium, this being the substrate of transversal axis within the wall of the BDs [42]. As represented in Figure 2, based on the progenitor segregation, multiple stem/progenitor cell niches locate in specific anatomical locations within adult human organs: biliary tree stem/progenitor cells (BTSCs) in PBGs along extrahepatic and large intrahepatic BDs, hepatic stem/progenitors (HpSCs) in or near CoH, pancreatic stem cells that appear confined to the biliary tree, particularly in glands within the hepatopancreatic common duct, and committed pancreatic progenitors are observed within the pancreatic ductal system, particularly in pancreatic duct glands [PDGs], which contain the early cells of the lineages in the pancreas (Figure 2) [37,43,44,45,46,47]. Detailed anatomical studies in humans have revealed a proximal-to-distal axis for the BTSC niche organization going from the proximal location (the hepatopancreatic ampulla), where the most primitive stem cells reside, to the distal location (the liver or the pancreas), where mature cells reside [45,47,48,49]. While observations in humans and experimental models clearly demonstrated that the replication of mature parenchymal cells ensures physiological turnover at homeostasis and after minor injuries in liver and pancreas, these stem/progenitor cell niches may contribute to repair pervasive, chronic, or massive injuries [45,47,48,49,50,51,52,53]. Interestingly, a parallel exists between embryological development processes and molecules and the strategies to differentiate both hHpSCs and hBTSCs in vitro toward determined mature fates [43,45,54]. Recapitulating the embryological development, appropriate hormonally defined medium have been tailored with respect to the native (or induced) lineage stage and desired fate (Figure 2).

## 2. Regenerative Medicine of the Liver

### 2.1. Controversy on Liver Regeneration

The involvement of bipotential hepatic stem/progenitor cells in liver regeneration has been challenged by the evidence on hepatocyte plasticity [55]. Specifically, the existence and role of stem/progenitor cells in the liver have been challenged by evidence from lineage-tracing studies in experimental models of liver diseases [56]. On the contrary, in human liver, the existence and role of the stem cell compartment have been largely supported by many studies in chronic liver diseases of different etiologies and in neoplastic transformation [57,58,59,60]. Finally, Forbes et al. have largely demonstrated that, in experimental models of liver injury, where hepatocyte senescence (as largely seen in human liver diseases) has been experimentally induced, the activation the stem/progenitor cell compartment clearly and significantly emerged [61]. Recently the controversy was further elegantly faced by the authors who, for the first time, used the single cell transcriptomic approach to create a normal human liver cell atlas [62]. Aizarani et al. individuated the EpCAM+ population as a strong candidate for potential involvement in homeostatic turnover, liver regeneration, and disease pathogenesis [62]. In this scenario, the recent article in *Nature* by Aizarani et al. adds further information in this debated topic, indicating that EpCAM+ population exhibits only stochastic expression of proliferation markers [62]. This is in accordance with studies on human normal liver tissue and strengthens the concept of a facultative progenitor compartment triggered to proliferate by regenerative needs due to pathological backgrounds [62].

### 2.2. State of Art of the Use of Hepatocytes and Stem Cells in Regenerative Medicine of the Liver

Orthotopic liver transplantation (OLT) represents the only curative treatment for acute liver failure and end-stage chronic liver disease [63]. However, OLT is limited by severe shortage of organ donors, and nearly 15% of patients die on the waiting list [49,64]. The rate of no transplanted patients for death or clinical deterioration among the patients on the OLT waiting list could be even underestimated. For these reasons, there is an urgency to find suitable and effective strategies to manage patients with advanced liver diseases when they are potentially treatable with OLT or when this treatment cannot be proposed. In this context, different strategies of cell therapy have been attempted and used not only for advanced cirrhosis but also for acute liver failure, fulminant hepatitis, inborn errors of metabolism, viral hepatitis, liver toxins, alcohol consumption, chronic cholestatic diseases, autoimmunity, and metabolic disorders such as non-alcoholic steato-hepatitis (NASH) [65]. Hepatocyte transplantation represents the proof of concept of live cell therapy. Indeed, clinical observations have demonstrated the safety of the procedure, and patients (~100) showed clinical improvement and/or partial correction of the underlying metabolic defect. However, in the majority of the cases, sustained benefits were not observed. The major challenges associated with hepatocyte transplantation are the limited supply of donor organs that are available to isolate good quality cells, the low cell engraftment, difficulties in cryopreservation, and the necessity of long-term immunosuppression [66,67,68,69,70,71,72,73,74,75]. Advanced grafting strategies have the potential to improve the outcome of the hepatocyte transplantation [66,75,76]. However, the improvement of regenerative medicine approaches for liver diseases requires the identification of sustainable and readily available cell sources [63]. Besides cell reprogramming, the use of MSCs or determined stem/progenitor cells isolated from adult or foetal human organs have been proposed for the regenerative medicine of the liver [49]. Adult or foetal stem cells have the advantage to require only minimal manipulation with respect to reprogrammed cells. Moreover, in some cases, they are also the physiological precursors of the mature cells of the target organ. MSCs are easily sourced, readily cryopreserved, and involve transplantation procedures with minimal, if any, complications [49]. Mesenchymal-derived stem cells are usually quiescent in the bone marrow and mobilized at times of injury. These comprise hematopoietic stem cells (HSCs), identifiable and isolatable from peripheral blood through the expression of CD34 and CD133, and mesenchymal stem cells (MSCs), which adhere to plastics, express CD105, CD73 and CD90, and differentiate to either osteoblasts, adipocytes or chondroblasts in vitro [49,77,78,79,80,81,82,83,84,85]. Transplantation of MSCs and hematopoietic stem cells into patients with liver disease has resulted in large numbers of clinical trials throughout the world [49,77,78,79,80,81,82,83,84,85]. The first published systematic review on this topic, comprising an evaluation of the quality of the studies, appeared in 2014 and included 33 papers reporting results from patients with advanced liver diseases (cirrhosis, severe steato-hepatitis, cirrhosis, and hepatocellular carcinoma candidates to surgical resection) [81]. Among these, only six randomized controlled trials (RCTs) were performed, and among them, only one was judged to be of good quality. Notably, this good quality RCT of autologous bone marrow mononuclear cell transplantation in patients with decompensated alcoholic liver disease by Spahr et al. resulted negatively [85]. A meta-analysis of the clinical outcomes of the transplantation of MSCs in the management of liver cirrhosis agreed upon the fact that the patient response occurred within days to weeks, but long-term effects (more than a few months) were not observed [65].

Citing the great expert Paolo Bianco [86], *“exogenous cueing of the directed differentiation of postnatal ‘mesenchymal’ stem cells is perhaps best seen as reprogramming (change of potential) rather than differentiation (expression of potential)”*. In fact, as far as the liver cell therapy is concerned, it has been clarified that the role of MSCs does not depend on the repopulation and regeneration determined by differentiation in liver parenchymal cells but by the production of factors and cytokines with multiple effects and by the expansion of macrophages concurrent with an upregulated expression of genes involved in inflammatory and regenerative pathways [49,77,78,79,80,81,82,83,84,85]. Interestingly, a single specific molecule secreted by the MSCs, the Milk Fat Globule-EGF Factor 8, protects against liver fibrosis in mice, as demonstrated recently by An et al. [77]. A recent multicentre phase-II open-label controlled trial of repeated autologous infusions of G-CSF mobilized CD133+ bone marrow stem cells in patients with advanced cirrhosis (versus conservative management or treatment with G-CSF alone) found no impact on liver function nor on fibrosis and the trial was interrupted also because complications of cirrhosis were even more common in the combined therapy group [79,82]. Finally, liver stiffness evaluation by transient elastography did not show any effect of the treatment. This study is of interest because of its rigorous design and evaluation, and as noted by Nicolas Lanthier, it highlights the importance of solid preclinical evidences in this field, the necessity to include a refined assessment of treatment response in future studies, and appropriate cell tracking experiments in humans to establish whether cells infused really do engraft the liver or not [87]. The possibility to perform investigations through liver biopsy samples in patients where this procedure may be proposed should be taken in high consideration, since, as demonstrated by Lanthier et al. [80], results within the tissues are highly informative, especially when clinical results are weak or absent. 

As far as pluripotent stem cells are concerned, they attract enormous interest worldwide. Generally, Embryonic Stem Cells (ESCs) evoke ethical concerns regarding their origin from human embryos, and they bear a clinical risk since their pluripotent nature makes undifferentiated ESCs capable to form malignant teratocarcinomas in vivo [64,88]. Although stem cells are usually hypoimmunogenic, non-autologous differentiated transplanted cells are challenged by adaptive immune responses such as local inflammation and rejection. The possibility of reprogramming adult somatic cells [88] attracted attention for the possibility to generate mature hepatocyte readily available for auto-transplantation [64]. In parallel, significant advancements have been obtained in the definition of protocols for the differentiation of human iPSC into functional mature hepatocytes (iPSC-Hep) [89], comprising obtaining mature hepatocytes either after partial reprogramming to pluripotency followed by directed hepatocyte differentiation (iMPC-Heps: induced multipotent progenitor cell-derived hepatocytes) [90] or directly reprogramming fibroblasts (or MSCs) by forcing the expression of hepatic transcription factors (induced hepatocytes/iHeps) [91,92,93]. The utilization of endodermal organ derived cells can be considered a major advancement, perhaps a more physiologic source than mesenchymal tissue derived cells, in liver reprograming attempts. In a recent paper by Wang et al., a set of defined small molecules plus mesenchymal feeder cells can covert human gastric epithelial cells into induced endodermal progenitors with the capability to differentiate into liver and pancreatic adult parenchymal cells [94]. Similarly, the clinical application of reprogrammed induced Pluripotent Stem Cells (iPSCs) raises concerns because the open possibility to undergo to uncontrolled tumorigenic expansion within the recipient [63,95]. Small molecule–based reprogramming has attracted much interest due to its safety and efficiency in controlling cell fates [96,97,98,99], but optimal protocols are not yet established, and reprogrammed cells lack sufficient maturity for widespread clinical use [63,76,100].

### 2.3. Foetal Liver as a Cell Source for Regenerative Medicine: Preclinical and Clinical Evidences

Hepatic stem/progenitors (HpSCs) reside in adult human liver and are located within the canals of Hering and bile ductules, where their potential to differentiate into functional mature hepatocytes and cholangiocytes can be stimulated, and new born mature cells can repopulate the specific diseased tissue [53,58,101]. In the foetal liver, HpSCs locate in the ductal plate and give rise to the generation of cholangiocytes of interlobular bile ducts and to periportal hepatocytes during liver development [38]. HpSCs can be easily isolate from human livers on the basis of the expression of specific surface antigens such as Epithelial Cell Adhesion Molecule (EpCAM) [52]. Moreover, a single Leucine-rich repeat-containing G-protein coupled receptor 5 (Lgr5)-positive liver stem cell can form organoids capable to be expanded ex vivo and to give rise to hepatocytes and cholangiocytes [50,102], maintaining a high stability at the chromosome and structural level [50]. HpSCs can be converted into functional hepatocytes in vitro and upon transplantation in vivo [50]. Therefore, these properties indicated HpSCs isolated from adult or foetal human livers as possible source for a cell therapy program. The biliary tree contains a large population of progenitor/stem cells named biliary tree stem/progenitor cells (BTSCs) [43,46,47]. Human BTSCs can be easily isolated from adult or foetal organs, comprising gallbladders, rapidly and clonal expand in culture, and can differentiate toward several endoderm cell types including hepatocytes, cholangiocytes, and pancreatic endocrine cells in vitro and in vivo [43,103]. Moreover, attempts to rescue an experimental model of fibrotic liver injury induced by carbon tetrachloride in mice through the transplantation of hBTSCs in mice resulted in a good amount of liver repopulation (average 10% of the liver mass) after a month from the infusion of 2 × 10^6^ hBTSCs in the portal circulation. This is associated with amelioration of serologic levels of transaminases and liver function indexes and with amelioration of the liver histology. Thus, in this context, the large availability of hBTSCs, especially from foetal livers, and their biological properties, comprising scarce or null immunogenicity and tumorigenicity [46,52], guaranty an ideal source of easily isolable and cultivable stem cells. Finally, both hHpSCs and hBTSCs can be easily produced under cGMP conditions, as demonstrated by our data [104]. Finally, hBTSCs, with respect to hHpSCs, can differentiate toward several endoderm cell types including hepatocytes, cholangiocytes, and pancreatic endocrine cells [43,105], being candidates for regenerative medicine for the liver and pancreas, including diabetes mellitus. Notably, hBTSCs express null or minimally HLA antigens and induce Fas-mediated apoptosis of activated T-cells [106]. Adult hBTSCs expressed class I HLA antigens, mostly A, C, E, F genes, and class II HLA antigens (DP, DQ, DR). Instead, Foetal hBTSCs expressed only class I HLA antigens. Accordingly, Rao et al. [107] described how EpCAM+ stem/progenitor cells isolated from foetal liver parenchyma exclusively express HLA class I. Lee et al. [108] found that adult hepatic stem/progenitor cells expressed MHC class I but not class II antigens. Moreover, hBTSCs constitutively express high levels of both mFas-L and sFas-L (adult > foetal hBTSCs) and are able to induce Fas-mediated apoptosis of activated CD4+ and CD8+ T-cells populations. The expression of Fas-L has been considered a key mechanism promoting homing and differentiation of transplanted hBMSCs [2]. Considering this literature, it appears that hBTSCs isolated from foetal liver are minimally immunogenic ideal cell sources. In clinical programs regarding cell therapy for liver diseases, transplantation of mature hepatocytes was associated with conventional immunosuppression regimes [5,6], while the transplantation of foetal liver stem cells in human patients does not need the administration of any immunosuppression regimen [7,8]. As far as issues concerning cGMP foetal liver cells are routinely isolated in cGMP. In our experience with this source, indeed, mothers underwent to assessment of infectious diseases (HCV, HBV, HIV, EBV, HEV, HDV, toxoplasmosis, rubella, cytomegalovirus, parvovirus, herpes simplex type 1 and 2, TPHA) as required by current regulation. The production of the cell is possible according to “the rules governing medicinal products in the European Union” and the European guidelines of GMP for medicinal products for human use (EudraLex—Volume 4 Good manufacturing practice Guidelines). Indeed, all media are disposable procured as “cGMP-manufactured” from commercial distributors. Sterility testing and endotoxin testing are readily highly available. The EpCAM sorting procedure is highly standardized and has been already used in program of human cell therapy without complication [104,109].

Clinical applications of ESCs evoke ethical concerns regarding their origin from human embryos and teratocarcinogenesis in vivo, while the clinical application of reprogrammed iPSCs raises concerns because the open possibility to undergo to uncontrolled tumorigenic expansion within recipient. Although autologous MSCs and HSCs are clinically safe, the recent negative results of RCTs limit future application of this strategy. It can be argued that the state of art of regenerative medicine of liver reached a point where further rigorous preclinical studies and high quality RCTs are required. These efforts could prove its efficacy. The foetal liver is the major candidate on the block. Human foetal liver comprises two stem cell niches—the ductal plates containing HpSCs, and PBGs containing BTSCs. HpSCs are facultative stem cells able to differentiate into functional mature hepatocytes and cholangiocytes, residing into the foetal human liver. To date, several case reports and a completed trial with HpSCs held in India by Habibullah and co-workers focused on patients affected by chronic liver disease of different etiologies, such as, biliary atresia, inborn errors of metabolism (Crigler-Najjar), non-alcoholic steato-hepatitis, viral cirrhosis, alcoholic cirrhosis, and drug toxicity. These anecdotal results suggest that human HpSCs can be effective in treating patients with liver disease [110]. More recently, a representative early publication by Habibullah and co-workers concerned a trial of 25 subjects and 25 controls with decompensated liver cirrhosis due to various causes. Subjects received foetal liver-derived EpCAM^+^ cell infusions into the liver via the hepatic artery. At a six-month follow-up, multiple diagnostic and biochemical parameters showed clear improvement, and there was a significant decrease in the patients’ MELD scores [110]. In western countries, an analogue clinical trial using cells isolated from foetal livers was started [106,111]. Indeed, Pietrosi et al. [112] treated nine patients affected by end-stage liver disease within intra splenic infusion of a total cell population obtained from the foetal liver and found that the procedure was safe and well tolerated with positive effects on clinical scores and on encephalopathy. Remarkably, immune suppression was not required, although donors and recipients were not matched for histocompatibility antigens.

hBTSCs are a possible source for stem/progenitor cells for liver regenerative medicine [37,44,113]. A key advantage is their ready availability from adult or foetal donors and from cholecystectomized patients in comparison with the scarcity of marginal livers usable for the isolation of adult HpSCs. Foetal stem/progenitor cells could also be ideal vehicles for delivering therapeutic genes into the liver [114,115,116]. Recently we reported preliminary results of a phase I/II clinical trial consisting in the via hepatic artery transplantation of human foetal biliary tree stem cells in patients with advanced cirrhosis [104]. We injected immune-selected EpCAM-positive cells which also co-expressed Leucine-rich repeat containing G protein coupled Receptor 5 (LGR5). In foetal livers, LGR5-positivecells were located in the ductal plate and in the epithelium of larger bile ducts [13]. In the gallbladder and hepatic common duct, surface epithelial cells and bud of PBGs, were diffusely positive for LGR5 [44]. Cell products were characterized by flow cytometry (FC) for EpCAM and LGR5 before transplantation. Estimated cell viability by trypan blue exclusion was routinely higher than 95%. The cells were immune-sorted by protocols in accordance with current good manufacturing practice (cGMP) and extensively characterized. Two patients with advanced cirrhosis (Child-Pugh C) have been submitted to the procedure and observed through a 12 months follow-up. The resulting procedure was found to be absolutely safe. Immuno-suppressants were not required, and the patients did not display any adverse effects correlated with cell transplantation or suggestive of immunological complications. From a clinical point of view, both patients showed biochemical and clinical improvement during the six-month follow-up, and the second patient maintained a stable improvement for 12 months. This report represents proof of the concept that the human foetal biliary tree stem cells are a suitable and large source for cell therapy of liver cirrhosis and represents the basis for an ongoing controlled clinical trial [104].

As widely demonstrated in hepatocyte transplantation, low cell engraftment is a major limitation to address in order to improve efficacy of liver cell therapy, even when it is based on stem cell populations. Cell therapy of liver diseases with human biliary tree stem cells (hBTSCs) may be biased by low engraftment efficiency. Several strategies targeting engraftment into the liver have been reported so far as. We have reported that coating the hBTSCs with hyaluronans (HAs), the primary constituents of all stem cell niches, could facilitate cell survival, proliferation, and, specifically, liver engraftment, given that HAs are cleared selectively by the liver [73,74,113]. HA coating could be translated into the clinic, given that GMP-grade HAs are already available for clinical use. HA cell coating markedly improved viability of hBTSCs after cryopreservation. HA-coating could be considered as a strategy for preconditioning for freshly isolated and cryopreserved hBTSCs in order to ameliorate outcomes and logistics of the transplantation of these cells in patients affected by severe liver diseases [73,74,113]. Finally, due to the difficulties to have sensitive clinical outcomes and because the recent negative clinical trials open several questions, cell labelling and tracking are pivotal tools to investigate further in order to improve our understanding of the biological mechanisms behind cell engraftment and verify the therapeutic effects of inoculated cells in vivo [74].

### 2.4. Organoids are Powerful Tool to Study Liver and Bile Duct Regeneration

Organoid culture is an advanced three-dimensional technology in which the cells can recapitulate the native physiology of cells in the healthy tissue of origin as well as the pathology of the cells in patient-derived tissue. This system allows the long-term expansion of cells in vitro and has been proposed as an in vitro disease-modelling tool for several genetic diseases. Recently, Clevers and Huch have developed the organoids culture for adult intestine [117], stomach [118], liver [102], and pancreas [50] of different species like mouse and human.

Huch and Clevers have also established a protocol for the long-term expansion and genetic manipulation of adult bile duct–derived bipotent progenitor cells in 3D condition [50]. Where the EpCAM negative cells were isolated as a purely ductal cell population and developed into organoid with high level of colony forming efficiency, the positive cells for marker EpCAM were sorted as a hepatocyte cells and formed no organoids. They have demonstrated the expanded cells were genetically stable and maintained normal chromosome numbers for more than three months [50]. Interestingly, this culture allows three months of expansion of liver organoids in vivo with 10-fold fewer base substitutions than iPS cell programming, which contain 1058–1808 de novo base substitution at only passage numbers between 15 and 25, compared to their parental somatic cells [119].

A recent study showed that human foetal liver organoids mimic human liver development, including several aspects of hepato-biliary organogenesis, and can differentiate toward hepatocyte and cholangiocyte structures with improved function [120].

Despite decades of basic research on biliary developmental biology, cholangiocyte pathophysiology, and the endogenous mechanism of biliary regeneration, the treatment of common bile duct disorders (CBD) is restricted to a few studies by the lack of biliary tissue [121]. Moreover, the biliary tree, given its vast distribution throughout the liver with distinct developmental origin and function, is the target for many of liver pathologies, so the more comprehensive study on biliary tree can be useful for treatment of both liver and biliary tree diseases. Recently, Fotios Sampaziotis et al. have shown the isolation and expansion of human cholangiocytes from extrahepatic biliary tree in the form of extrahepatic cholangiocyte organoids [122]. In this study, co-expression of CK7 and CK19 following 20 passages in culture and other biliary markers like a hepatocyte nuclear factor-1β (HNF-1β), g-glutamyl transferase activity (GGT), secretin receptor (SCTR), sodium-dependent bile acid transporter (SLC10A2; also known as ASBT), CFTR, and SRY-box 9 (SOX9) were established by RT–qPCR and immunofluorescence (IF) analyses. They also demonstrated the engraftment of Cholangiocyte organoids in mice after 12 weeks expressing biliary markers such as CK19 [122].

Therefore, organoid culture derived from bile ducts and gallbladder can be a very relevant system to study human biology, physiology, regeneration, and the development of the biliary tree and even the liver. In addition, 3D organoid culture enables the study of molecular mechanisms driving pathologies and at the same time can provide potential tools to manipulating genomes and facilitate gene correction for regenerative medicine and autologous therapy. Researchers have expanded organoids from A1AT-deficiency patients in vitro when it recapitulated the pathology in vivo. Similarly, organoids from Alagille syndrome patients mimicked structural duct defects of these patients and presented the genetic disorders of this pathology [50]. The ease of genetic manipulation using either retroviruses or CRISPR/Cas9 combined with 3D culture provides a powerful tool for comprehensive understanding of gene functions and genetic modification. It also might facilitate and accelerate advancement in the field of personized medicine for biliary and liver disease.

## 3. Conclusions

Since organoids replicate key structural and functional features of the corresponding in vivo organ, their stem cells of origin acquire phenotype and functions of mature cells. Thus, the application of organoids to regenerative medicine highlights the need to face the question of whether, among mature versus stem/progenitor cells, a determined cell source is better suitable for a specific clinical need. Some years ago, Forbes and Newsome presented a very conveyable paradigm according to which the different available cell sources should be tailored for the specific pathological conditions according to the specific and peculiar properties they are endowed with [81]. There is no conclusive evidence on the applicability of this paradigm since most of the clinical trial results were negative. The existence of multiple sources with multiple properties and the occurrence of different and variable clinical settings (liver cirrhosis, acute on chronic liver failure, acute liver failure, inborn errors of metabolism) require us to envision the new figure of the cell therapist hepatologist.

The state of art of regenerative medicine of liver reached a point where further rigorous preclinical studies and high quality RCTs are required. These efforts could determine a cell source when this will finally prove its efficacy. Foetal liver is the major candidate on the block. The foetal liver possesses a unique feature given the co-existence of endodermal and mesenchymal derived cells and is hypothetically the more useful and qualified largely available source to address the main areas (fibrosis remodeling and liver repopulation) required for an effective cure of liver cirrhosis. Moreover, it contains a population of pluripotent stem cells, the hBTSCs, and thus is the unique highly available source candidate contemporarily for the regenerative medicine of the liver and pancreas. However, researchers and clinical investigators in regenerative medicine of the liver should adopt rigorous clinical studies for the foetal liver, as done only recently concerning the use of autologous MSCs in cirrhotic patients. Moreover, preclinical studies should be solid and tailored to address the questions raised by the clinical trials. Importantly, repopulation of the liver and the proliferation and differentiation of transplanted cells should be investigated systematically in the tissues along different and long-term time points. Different experimental models should be evaluated to investigate specific etiopathogenetic features that may impact the outcomes of the cell therapy. For example, models of liver fibrosis are best candidate to study intrahepatic factors associated with the interactions and effects of transplantation of exogenous cells, while NAFLD/NASH models may reveal potential systemic factors impacting on the effects of exogenous cells transplanted into the liver.

## Figures and Tables

**Figure 1 cells-08-00914-f001:**
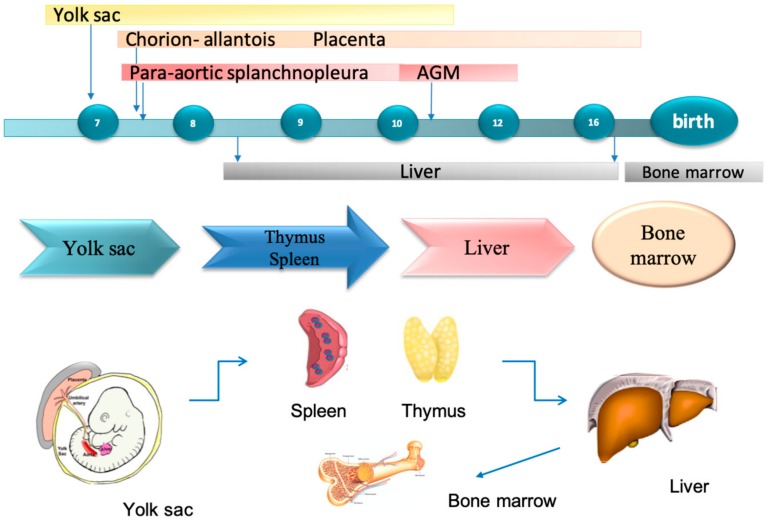
Timeline of fetal hematopoiesis. Primitive hematopoietic cells appear in the yolk sac for seven weeks. Hematopoiesis is then temporarily shared between the liver, spleen, and thymus. After blood circulation starts, primitive hematopoietic cells get into the circulation and mature. At 10 weeks, hematopoietic progenitor cells gradually migrate from the aorta-mesonephros-gonad (AGM) region to colonize the liver, which becomes the major hematopoietic organ. Hematopoiesis shifts from foetal liver to bone marrow at 16 gestational weeks.

**Figure 2 cells-08-00914-f002:**
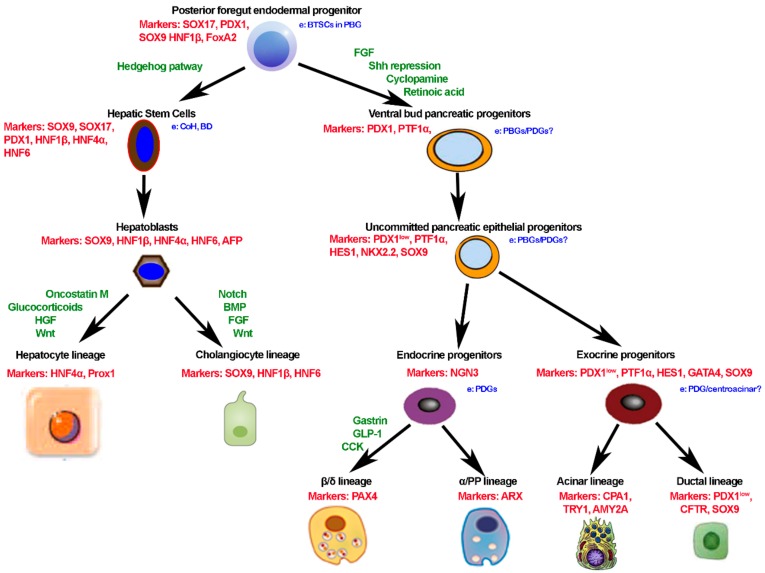
Maturation of hepatic, biliary, and pancreatic lineages from foregut endoderm progenitor cells. In the figure we recapitulate the embryological development, indicating key biomarkers and stimuli to induce cell differentiation. Abbreviations (in order of appearance): a, adult; e, embryonic; BTSCs, biliary tree stem/progenitor cells; PBGs, peribiliary glands; PDGs, pancreatic ductal glands; CoH, canal of Hering; BD, bile duct; AFP, alpha-fetoprotein; AMY2A, amylase 2A; CPA1, carboxypeptidase A1; SOX, sex determining region Y-box; PDX1, pancreatic and duodenal homebox 1; HNF, hepatocyte nuclear factor; Fox, forkhead box; PTF, pancreas transcription factor-like; HES1, transcription factor HES1; NGN, neurogenin; GATA, erythroid transcription factor; PAX, paired box; ARX, aristaless related homeobox; TRY, transcription factor TRY; CFTR, cystic fibrosis transmembrane conductance regulator; CPA, carboxypeptidase; FGF, fibroblast growt factor; Shh, sonic hedegehog; HGF, hepatocyte growth factor; BMP, bone morphogenetic protein; GLP-1, glucagon like peptide 1; IGF-1, insulin-like growth factor 1.

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
