# Peer review of "Functions and the Emerging Role of the Foetal Liver into Regenerative Medicine"

_cells, 2019, doi:10.3390/cells8080914_

Round 1

Reviewer 1 Report

This nice review is well-organized and provides valuable information for the researchers in this field. In this review, the authors reviewed the embryogenesis, maturation of hepatic lineages from stem cells, and regeneration medicine of the liver. This review article can be accepted.  

Reviewer 2 Report

Giancotti et al is a well written review paper.  There is one area that I recommend the authors to clarify.  The authors in one place discussed about the so called reprogrammed somatic cells as the potential cell source for regenerative medicine.  In the other places, the authors are talking about progenitor cells.  What is the difference between reprogrammed somatic cells and progenitor cells?  Is there any basis of somatic cell reprogramming?  Does this concept suggest the there is no progenitor cells in the liver?  I think the authors need to discuss about this controversy.  In addition, 3D organoid is a unit that include most differentiated cells.  For the regenerative medicine, would it be more attractive to transplant progenitor or adult stem cells?  The authors need to discuss both con and pro of this system.

Author Response

Giancotti et al is a well written review paper.  There is one area that I recommend the authors to clarify.

The authors in one place discussed about the so called reprogrammed somatic cells as the potential cell source for regenerative medicine.  In the other places, the authors are talking about progenitor cells. What is the difference between reprogrammed somatic cells and progenitor cells? Is there any basis of somatic cell reprogramming? 

Reply: We thank the referee for the constructive comments. In this paper indeed we have presented all the possible candidate cell sources in liver cell therapy. By doing so we have included both pre-clinical and clinically tested sources. Obviously reprogrammed somatic cells have been never tested in human for several reasons already discussed within the current manuscript (see page 9, lines 334-342), while, tissue stem/progenitor cells like bone marrow derived stem cells (hematopoietic and mesenchymal stem cells), and hepatic stem/progenitor cells have been already tested in clinical trials in humans (see page 8, lines 278-298).

Does this concept suggest the there is no progenitor cells in the liver?  I think the authors need to discuss about this controversy.

The involvement of bipotential progenitor cells in liver regeneration has been challenged by the evidence on hepatocyte plasticity [93]. Specifically, the existence and role of stem/progenitor cells into the liver have been challenged by evidences from lineage tracing studies in experimental models of liver diseases [94]. On the contrary, in human liver the existence and role of the stem cell compartment have been largely supported by many studies in chronic liver diseases of different aetiologies and in neoplastic transformation [95-98]. Finally, Prof. Forbes and his associates have largely demonstrated that in experimental models of liver injury where hepatocyte senescence (as largely seen in human liver diseases) has been experimentally induced, then the activation the stem/progenitor cell compartment clearly and significantly emerged [99]. Recently the controversy was furtherly elegantly faced by Authors who used a single cell transcriptomic approach to create a normal human liver cell atlas [100]. Aizarani et al. individuated the EpCAM+ population as a strong candidate for potential involvement in homeostatic turnover, liver regeneration, and disease pathogenesis. In this scenario, this article adds further information in this debated topic; interestingly, authors indicated that EpCAM+ population exhibits only stochastic expression of proliferation markers; this is in accordance with studies on human normal liver tissue and strengthen the concept of a facultative progenitor compartment triggered to proliferate by regenerative needs due to pathological backgrounds [100]. As suggested by the Referee we have added a discussion about the above mentioned controversy within the manuscript (page 9, lines 343-359; page 17, lines 752-753; page 18, lines 754-767).

In addition, 3D organoid is a unit that include most differentiated cells. For the regenerative medicine, would it be more attractive to transplant progenitor or adult stem cells?  The authors need to discuss both con and pro of this system.

Reply: we thank the Referee for the constructive criticism.

Organoids can be derived from two types of stem cells: either (1) pluripotent stem cells (PSCs) including embryonic stem cells (ESCs) and induced pluripotent stem cells (iPSCs) or (2) organ-specific adult stem cells (ASCs), which are tissue-specific resident stem cells [123]. However, when grown in a 3D environment, these stem cells self-organize into organoids that replicate key structural and functional features of the corresponding in vivo organ, thus acquiring phenotype and functions of mature cells, as correctly stated by the referee. Thus, the application of organoids to regenerative medicine highlights the need to face the question whether among mature vs stem/progenitor cells a determined source is better suitable for a specific clinical need. In liver regenerative medicine, this hot topic has been faced regardless organoids or single cells are concerning. The answer is not always the same as elegantly highlighted by Forbes and Newsome which presented a very conveyable paradigm according which the different available cell sources should be tailored for the specific pathological conditions according the specific and peculiar properties they are endowed with [81]. There are not conclusive evidences about this issue yet. We now propose in the manuscript conclusion our personal opinion according which, the existence of multiple sources with multiple properties, and the occurring of different and variable clinical settings (liver cirrhosis, acute on chronic liver failure, acute liver failure, inborn errors of metabolism),  request to envision a new figure in medicine; the cell therapist hepatologists which should be capable to define the clinical setting carefully, the expected prognosis, the more appropriate cell therapies (multiple or mixed sources), the best timing (and eventually the re-treatment), and finally the expected results, and design and conduct statistically and methodological valid clinical trial (see page 12, lines 517-535; page 19, lines 810-811).